# Gate-tunable plasmons in mixed-dimensional van der Waals heterostructures

Sheng Wang [1,2,11✉], SeokJae Yoo [1,3,11✉], Sihan Zhao[1,11], Wenyu Zhao[1], Salman Kahn [1], Dingzhou Cui[4], Fanqi Wu[4], Lili Jiang[1], M. Iqbal Bakti Utama[1,2,5], Hongyuan Li [1,6], Shaowei Li [1,2], Alexander Zibrov[1,2], Emma Regan [1,2,6], Danqing Wang[1,2,6], Zuocheng Zhang[1], Kenji Watanabe [7], Takashi Taniguchi [8], Chongwu Zhou[4,9] & Feng Wang [1,2,10✉]

Surface plasmons, collective electromagnetic excitations coupled to conduction electron oscillations, enable the manipulation of light–matter interactions at the nanoscale. Plasmon dispersion of metallic structures depends sensitively on their dimensionality and has been intensively studied for fundamental physics as well as applied technologies. Here, we report possible evidence for gate-tunable hybrid plasmons from the dimensionally mixed coupling between one-dimensional (1D) carbon nanotubes and two-dimensional (2D) graphene. In contrast to the carrier density-independent 1D Luttinger liquid plasmons in bare metallic carbon nanotubes, plasmon wavelengths in the 1D-2D heterostructure are modulated by 75% via electrostatic gating while retaining the high figures of merit of 1D plasmons. We propose a theoretical model to describe the electromagnetic interaction between plasmons in nanotubes and graphene, suggesting plasmon hybridization as a possible origin for the observed large plasmon modulation. The mixed-dimensional plasmonic heterostructures may enable diverse designs of tunable plasmonic nanodevices.

[1] Department of Physics, University of California at Berkeley, Berkeley, CA, USA. [2] Materials Sciences Division, Lawrence Berkeley National Laboratory, Berkeley, CA, USA. [3] Department of Physics, Korea University, Seoul, Korea. [4] Department of Chemical Engineering and Materials Science, University of Southern California, Los Angeles, CA, USA. [5] Department of Materials Science and Engineering, University of California at Berkeley, Berkeley, CA, USA. [6] Graduate Group in Applied Science and Technology, University of California at Berkeley, Berkeley, CA, USA. [7] Research Center for Functional Materials, National Institute for Materials Science, Tsukuba, Japan. [8] International Center for Materials Nanoarchitectonics, National Institute for Materials Science, Tsukuba, Japan. [9] Department of Electrical Engineering, University of Southern California, Los Angeles, CA, USA. [10] Kavli Energy NanoScience Institute at the University of California, Berkeley and the Lawrence Berkeley National Laboratory, Berkeley, CA, USA. [11] These authors contributed equally: Sheng Wang, SeokJae Yoo, Sihan Zhao. ✉email: shengwang16@berkeley.edu; seokjaeyoo.nano@gmail.com; fengwang76@berkeley.edu

Surface plasmons are collective charge oscillations coupled to electromagnetic waves[1]. Dimensionality has profound effects on the plasmon properties. Plasmons in low dimensions have been intensely studied for their potential in manipulating light–matter interactions far below the diffraction limit[2–5]. Notably, in one-dimensional (1D) materials such as single-walled carbon nanotubes (SWNTs), electrons are strongly correlated with each other as a Luttinger liquid, resulting in peculiar properties of 1D Luttinger plasmons[6–8]. The 1D Luttinger plasmon in metallic SWNTs combines a nondispersive plasmon velocity, deep subwavelength confinement, and low loss, but it cannot be actively modulated by gating due to its intrinsic independence on carrier density[6,7,9]. Two-dimensional (2D) graphene sheets encapsulated between hexagonal boron nitride (h-BN) layers host highly confined and low-loss 2D plasmons with gate tunability[10–16]. Despite extensive efforts to investigate surface plasmons in each dimension as well as in coupled plasmonic systems within the same dimension (that is, 3D-3D, 2D-2D, and 1D-1D), the exploration of plasmonic modes in systems of mixed dimensionality has been surprisingly limited[6,8,10,11,17–21]. Recently, mixed-dimensional van der Waals (vdW) heterostructures have emerged as a promising platform to explore intriguing physical phenomena and new device functionalities[22–25]. Theoretical studies further suggest that the coupling between plasmonic materials of mixed dimensionality can drastically change the plasmon dispersion[18,26,27]. However, an experimental realization of the collective plasmon modes of a coupled mixed-dimensional heterostructure has yet to be demonstrated.

In this article, we report experimental and theoretical studies of plasmons in mixed-dimensional SWNT/h-BN/graphene heterostructures, which serve as an exemplary 1D-2D hybrid plasmonic system. The results indicate that the experimentally observed tunable plasmon modes are most likely to be interpreted as hybrid plasmons due to the coupling between the 1D SWNT plasmons and the 2D graphene plasmons. This coupling is theoretically expected to lead to efficient electrical control of the highly localized plasmon excitations along the nanotube. Our findings demonstrate that dimensionally mixed coupled plasmonic systems may host plasmons with on-demand properties that cannot be attained in the constituents alone. The approach can be used to design hybrid plasmonic devices with varying functionalities.

## Results

### Fabrication of mixed-dimensional van der Waals heterostructure

As schematically shown in Fig. 1a, the designed heterostructure has a top-down layout of SWNT/top h-BN/ graphene/bottom h-BN/SiO$_2$/Si. A thin layer of top h-BN (~2 nm) is inserted between SWNT and graphene to avoid direct charge transfer while maintaining efficient electromagnetic coupling. Ultraclean SWNTs are directly grown onto SiO$_2$/Si substrates by chemical vapor deposition (CVD)[8,9]. Graphene and h-BN flakes are mechanically exfoliated onto SiO$_2$ (285 nm)/Si substrates. SWNT/h-BN/Graphene heterostructures are assembled by the standard polymer stamp dry-transfer technique[28] (see Methods for details). The top h-BN flake does not fully cover the graphene to allow a direct metal contact with the exposed graphene region. The optical image of an as-prepared heterostructure is shown in Fig. 1b. The graphene and top h-BN layers are respectively outlined by black and green dashed lines. The overlapping region between graphene and top h-BN with SWNTs (optically invisible) on top of it constitute the SWNT/h-BN/graphene heterostructure.

### Infrared nano-imaging of gate-tunable plasmons

We probe the plasmon modes in the heterostructure using infrared scanning near-field optical microscopy (IR-SNOM) as illustrated in Fig. 1a[7–11]. This infrared nano-imaging technique is based on a tapping mode atomic force microscopy (AFM). The metallic AFM tip is illuminated with a focused infrared beam of wavelength 10.6 μm (28.3 THz). The sharp tip (~20 nm radius at the apex) acts as an optical nanoantenna and concentrates the incident light into a near-field nanoscale spot, which launches the plasmons in the sample. The launched plasmon wave propagates towards and reflects off the sample edges or other scatterers, forming a standing wave with a period of half plasmon wavelength, $\lambda_P/2$. The signal backscattered by the tip apex contains essential plasmon information and is captured by a mercury cadmium telluride (MCT) detector in the far field. By a raster scan of the sample surface, near-field images are obtained simultaneously with topography. Figure 1c, d display the topography and corresponding near-field image of a representative heterostructure area. Carbon nanotubes labeled M1 and M2 in Fig. 1c are metallic SWNTs with bright contrast in the near-field image due to tip-launched plasmons whereas the nanotube labeled S is a semiconducting SWNT with a negligible near-field response due to the lack of free electrons[8].

Figure 2 presents the main results of infrared nano-imaging of plasmons in an SWNT/h-BN/graphene heterostructure at a set excitation wavelength $\lambda_0$ of 10.6 μm. The topography and corresponding near-field images of a long SWNT M1 are shown in Fig. 2a–i. From the height profile in the topography image in Supplementary Fig. 1a, the diameter of SWNT M1 is determined to be 1.5 nm. When the gate voltage $V_g$ is varied away from the charge neutral point $V_{cnp}$, the graphene carrier density $n$ and associated Fermi energy $E_F$ are tuned continuously as $n = C_g|V_g - V_{cnp}|$ and $E_F = \hbar v_F\sqrt{\pi n}$, where $C_g$ is the capacitance density, $\hbar$ is the reduced Planck constant, and $v_F$ is the Fermi velocity[10–12,14,15]. We study the heterostructure for a wide range of gate voltage with a maximum at $-120$ V, corresponding to a hole density maximum of $7.6\times 10^{12} cm^{-2}$ and a Fermi energy maximum of $-0.26$ eV (see Supplementary Note 3 for details). As the carrier density is increased, two fringes parallel with the nanotube in the near-field images become more visible and separated. Here the nanotube acts as an effective 1D plasmon line reflector of the 2D graphene plasmon. The twin fringes arise from the interference between the tip-launched and nanotube-reflected graphene plasmon waves[22]. The evolution of the twin fringes with gate voltage indicates that the graphene carrier density and associated graphene plasmons are effectively tuned by the applied gate voltage $V_g$ in our measurement. More importantly, we clearly observe that prominent near-field oscillations emerge near the nanotube ends and that they depend sensitively on the gate voltage. The oscillation peaks near the right end correspond to the constructive interference between the plasmon wave excited by the tip and that reflected by the right end[6,7]. Note that the plasmon wave reflected by the left end experiences substantial damping during the long-distance roundtrip propagation and therefore has negligible contribution to the signal near the right end. The white double arrow (Fig. 2d, i) denotes the plasmon wavelength, which is equal to twice the oscillation period. It's seen that the plasmon wavelength becomes longer as the gate voltage is varied further away from 0 V. A similar trend can also be observed at the nanotube's left end but is less discernible than at the right end. We believe that impurity/defect-induced plasmon scattering near the left end complicates the plasmon interference patterns.

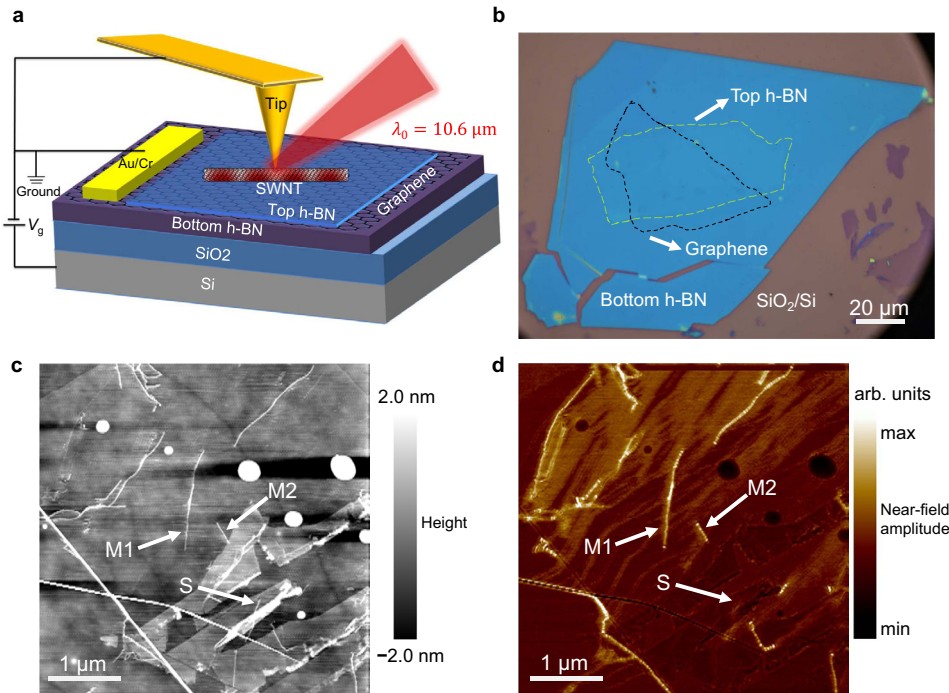

**Fig. 1 Infrared nano-imaging of plasmons in a SWNT/h-BN/graphene heterostructure. a** Schematic of infrared nano-imaging of plasmons in a SWNT/h-BN/graphene heterostructure. The designed heterostructure is assembled by the standard polymer stamp dry-transfer technique with a top-down layout of SWNT/top h-BN/graphene/bottom h-BN/SiO$_2$/Si. Graphene carrier density can be electrostatically tuned by applying a gate voltage $V_g$ between the Au/Cr electrode contacting the exposed graphene and the conductive Si layer. For infrared nano-imaging, a gold-coated AFM tip is illuminated with a focused infrared light beam of wavelength 10.6 μm. The backscattered light from the tip-sample system is collected to extract the near-field signal. **b** Optical image of an as-prepared heterostructure. The borders of graphene and top h-BN layers are respectively outlined by black and green dashed lines for clarity. The overlapping region between graphene and top h-BN with SWNTs (optically invisible) on top of it constitutes the SWNT/h-BN/graphene heterostructure. **c**, **d** Topography and corresponding near-field image of a representative heterostructure area. M1 and M2 in **c** are metallic SWNTs with bright contrast in the near-field image due to plasmon excitations whereas S in **c** is a semiconducting SWNT with a negligible near-field response due to the lack of free electrons.

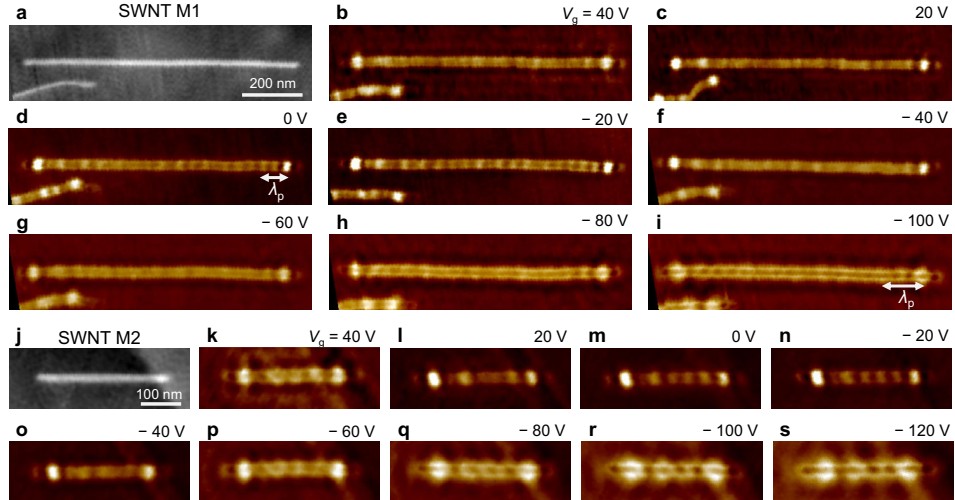

**Fig. 2 Gate-tunable plasmons in a SWNT/h-BN/graphene heterostructure. a** Topography of a long SWNT M1. **b–i** Corresponding near-field images of SWNT M1 at various gate voltages from 40 to −100 V. The twin fringes parallel with the nanotube arise from the interference between the tip-launched and nanotube-reflected graphene plasmon waves. As the gate voltage is increased, the twin fringes become more visible and separated. This evolution shows that the graphene carrier density and associated graphene plasmons are continuously tuned by the applied gate voltage. We clearly observe that prominent near-field oscillations emerge near the right nanotube end and that they depend sensitively on the gate voltage. The plasmon wavelength $\lambda_p$, equal to twice the oscillation period in the near-field images, is marked by the white double arrow (**d** and **i**) and becomes longer as the gate voltage is varied further away from 0 V. **j** Topography of a short SWNT M2. **k–s** Corresponding near-field images of SWNT M2 at various gate voltages from 40 to −120 V. SWNT M2 acts as a Fabry–Perot plasmonic nanocavity where propagating plasmons are reflected back and forth by both ends and produce a collective response. From **m–s**, the number of antinodes decreases from 7 to 4. The achieved plasmon wavelength modulation is estimated to be ~75%.

The modulation of plasmon excitations is also evident in a short SWNT M2. Fig. 2j–s show the topography and corresponding near-field images of SWNT M2 at various gate voltages. The diameter of SWNT M2 is determined to be 1.5 nm from the height profile in the topography in Supplementary Fig. 1b. SWNT M2 acts as a Fabry–Perot plasmonic nanocavity where propagating plasmons are reflected back and forth by both ends and produce a collective response[9]. With increasing graphene carrier density (Fig. 2m–s), we clearly observe a decreasing number of near-field signal maxima along the nanotube. The longitudinal cavity modes of the surface plasmon can be described by $2k_pL + 2\emptyset_R = 2\pi l$, where $k_p = 2\pi/\lambda_p$ is the plasmon momentum, $L$ is the length of the cavity, $\emptyset_R$ is the effective reflection phase shift at one end, and $l$ is the resonance order. As $V_g$ is increased to the negative side (Fig. 2m–s), the cavity resonance order $l$ decreases and the number of antinodes (that is, intensity maxima) decreases from 7 to 4. The achieved plasmon wavelength modulation can be estimated to be ~$(1/4 - 1/7)/(1/7)$ ~75%. The gate-tunable plasmons in SWNT M1 and M2 at the positive gate voltage side are shown in Supplementary Fig. 8 and show largely symmetric behavior compared with the negative side.

As we dope graphene by increasing $|V_g|$, the intensity maxima move from the nanotube axis to the top and bottom edges (Fig. 2m–s) owing to the coupling between SWNT and graphene plasmons. This behavior complicates the plasmon interference pattern and accurate determination of the plasmon wavelength. To determine the plasmon wavelength from SWNT near-field images reliably, we employ an image processing algorithm which enables robust local peak detection in complex data (see Methods for details). The intensity maxima in 2D near-field images (Fig. 2) are found algorithmically and subsequently projected onto the SWNT axis to obtain the peak-to-peak distances, providing the wavelength of the plasmon mode. The uncertainties of the determined plasmon wavelengths are described by the standard deviation of the averaged peak-to-peak distances in plasmon oscillations along SWNTs. Plasmon wavelengths determined from both SWNT M1 and M2 near-field images are summarized in Fig. 3b and show consistent monotonic growth from ~80 to ~140 nm with increasing $|V_g|$ from 0 to 120 V.

The gate-modulation of plasmon wavelength represents possible evidence for the dimensionally mixed coupling between 1D plasmons in SWNTs and 2D plasmons in graphene. Metallic SWNTs host 1D ultracompact and low-loss Luttinger liquid plasmons. At an excitation wavelength $\lambda_0$ of 10.6 μm, the bare SWNT plasmon has a carrier density-independent plasmon wavelength of $\lambda_{p,SWNT}$ ~84 nm as illustrated by the black dotted line in Fig. 3a[6,7]. On the other hand, the graphene plasmon wavelength $\lambda_{p,gr}$ depends on the carrier density[10–12,14] (black dashed line in Fig. 3a). In the nonretarded regime $(\lambda_0 \gg \lambda_{p,gr})$, graphene plasmon features a dispersion relation, $k_{p,gr} = 2i\omega\varepsilon_0\varepsilon_{med}/\sigma(\omega, E_F)$, where $\varepsilon_0$ is the vacuum permittivity, $\varepsilon_{med}$ is the relative permittivity of the surrounding medium, $\sigma(\omega, E_F)$ is the graphene optical conductivity at an excitation frequency of $\omega$, and a Fermi energy of $E_F$. As shown in Fig. 3a, when $2|E_F|$ is less than the plasmon energy $E_{p,gr}$ (that is, $|E_F| < 0.07$ eV), the graphene plasmons suffer from severe Landau damping by exciting interband electron-hole pairs. When the intraband conductivity dominates at substantially high Fermi energy, the graphene plasmon wavelength $\lambda_{p,gr} = 2\pi/Re(k_p)$ can be written as $\lambda_{p,gr} = (e^2|E_F|)/(\varepsilon_0\varepsilon_{med}\hbar^2\omega^2)$, where $Re(k_p)$ is the real part of complex-valued $k_p$. This $E_F$-dependence is also clearly observed in the infrared nano-imaging of the gate-dependent graphene plasmon interference patterns near the edge of graphene in the same heterostructure (see Supplementary Note 3 for details). The crossing of the bare SWNT and graphene plasmon curves occurs at a graphene Fermi energy of $|E_F|$ ~0.16 eV. This crossing is split by the optical coupling between 1D SWNTs and 2D graphene when they are stacked together, implying the possible emergence of two-hybrid plasmon modes. Our data show that the plasmon excitations change gradually over $E_F$ near charge neutrality in our heterostructure (Fig. 2). This behavior rules out an alternative picture based on the reflection of graphene plasmons by SWNT because graphene plasmons are strongly suppressed and not observable at the low-doping regime, that is $|E_F| < 0.12$ eV or $|V_g| < 30$ V.

**Theoretical predictions.** A theoretical model is established to better understand the large plasmon modulation in this mixed-dimensional heterostructure, suggesting the likely origin to be the

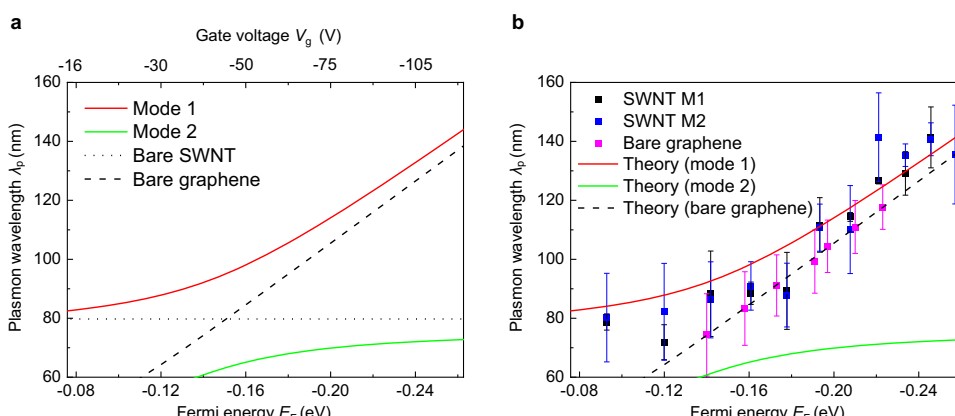

**Fig. 3 Possible evidence for the emergence of hybrid plasmons in the SWNT/h-BN/graphene heterostructure. a** Theoretically predicted plasmon wavelength of the hybrid plasmon modes of the coupled SWNT/h-BN/graphene heterostructure. The crossing of bare 1D SWNTs (black dotted line) and 2D graphene plasmons (black dashed line) is split into two-hybrid plasmon modes (red and green solid lines) by the strong optical coupling between the plasmon modes. The plasmon wavelengths of bare SWNTs (black dotted line) are obtained by the mode 1 at $E_F = 0$ eV to include the screening effect of the undoped graphene. **b** Experimentally extracted gate-dependent plasmon wavelengths from near-field images of SWNT M1 and M2 are represented by the black and blue squares, respectively. The error bars represent the standard deviation of the averaged peak-to-peak distances in plasmon oscillations along SWNTs. Theoretical predictions are plotted for comparison (red and green solid lines). The experimental plasmon wavelength dependence on Fermi energy is well reproduced by the theoretical upper plasmon branch (mode 1) of the coupled system. The experimental and theoretical bare graphene plasmon wavelengths versus Fermi energy are respectively represented by pink squares and black dashed lines, showing different trends compared with the upper hybrid plasmon mode 1. Source data are provided as a Source Data file.

plasmon hybridization. SWNTs can be modeled as ultrathin cylindrical waveguides with radius $R$ and relative permittivity $\varepsilon_{in}$ embedded in a medium with relative permittivity $\varepsilon_{med}$, where plasmons propagate along the axial direction in SWNTs. The axial plasmon propagation is accompanied by strong field localization in the radial direction. When graphene is placed in the vicinity of SWNT at a distance $h$, the near field of SWNT plasmons can excite graphene plasmons, which in turn alters the radial field confinement of SWNT plasmons. The strongly perturbed radial fields result in the strong modification of the axial plasmon propagation, that is the coupling between two plasmons, eventually yielding two hybrid modes. To describe this optical coupling mechanism analytically, we solve the electromagnetic boundary condition problem at the SWNT surface in the SWNT/h-BN/graphene heterostructure. Using a resonant momentum approximation by keeping the dominant contribution of the 1D plasmonic field reflected by graphene in its angular spectrum (see Supplementary Note 5 for details), we can obtain the hybrid plasmon dispersion relation for the SWNT/h-BN/graphene heterostructure,

$$\frac{\varepsilon_{in}}{\varepsilon_{med}}\frac{k_{\rho,med}}{k_{\rho,in}} = -\frac{I_0\left(k_{\rho,in}R\right)K_1\left(k_{\rho,med}R\right) - r\left(k_{\rho,med},E_F\right)e^{-k_{\rho,med}(2h-R)}/2}{I_1\left(k_{\rho,in}R\right)K_0\left(k_{\rho,med}R\right) + r\left(k_{\rho,med},E_F\right)e^{-k_{\rho,med}(2h-R)}/2},$$

where the radial momenta inside and outside SWNT are defined by $k_{\rho,in/med} \equiv \sqrt{k_p^2 - k_{in/med}^2}$ and $I_n$ and $K_n$ are the $n$-th order modified Bessel functions of the first and the second kind, respectively. $r(k, E_F)$ is the gate-tunable Fresnel-like graphene response function for the incoming evanescent waves of an arbitrary momentum $k$ larger than $k_{med}$. To solve Eq. 1 analytically, we use the permittivity ratio $\varepsilon_{in}/\varepsilon_{med} = -210 + 23.52i$ to include the substrate effect. It yields $\lambda_p$ ~80 nm and $Q$ ~15 at $E_F = 0$ eV, which are consistent with the experimental observations in Figs. 2 and 3. Further details in the theoretical model and its derivation can be found in Supplementary Note 5.

Solving Eq. 1 for the complex-valued plasmon momentum $k_p$ at a given Fermi energy $E_F$, we can obtain the plasmon wavelength $\lambda_p$ and the quality factor $Q$ of the hybrid plasmon modes using the relation $k_p = (2\pi/\lambda_p)(1 + i/Q)$. The hybrid plasmon dispersion relation (Eq. 1) in the limit $r(k, E_F) = 0$ reduces to that of freestanding SWNTs providing a single solution. On the contrary, Eq. 1 with $r(k, E_F) \neq 0$ yields double solutions exhibiting the plasmon hybridization in the mixed-dimensional heterostructure as shown in Fig. 3a (red and green lines). Theoretically predicted plasmon dispersions in Fig. 3a differ from the dispersions of the bare SWNT plasmons and the bare graphene plasmons. As illustrated in Fig. 3a, plasmon mode 1 (mode 2) behaves as SWNT-like (graphene-like) at smaller $V_g$, but progressively becomes graphene-like (SWNT-like) at higher $V_g$. The experimentally obtained plasmon wavelength as a function of gate voltage from infrared nano-imaging data of both SWNT M1 and M2 are displayed in black and blue squares, respectively as illustrated in Fig. 3b. The dependence is well reproduced by the theoretical upper plasmon branch (mode 1, red solid line) of the coupled system. The experimental and theoretical bare graphene plasmon wavelength versus Fermi energy are respectively represented by pink squares and black dashed lines, showing different trends compared with the upper hybrid plasmon mode 1. The lower plasmon branch (mode 2, green solid line) is predicted by the analytic results but is not clearly resolved in the infrared nano-imaging data.

We hypothesize that the absence of the clear lower plasmon mode in the experimental data is due to a combination of two effects. First, for the lower plasmon mode, the out-of-phase

charge oscillations in SWNT and graphene leads to a shorter plasmon wavelength and concentration of the electric field at the gap between SWNT and graphene, resulting in the weaker coupling of this mode to the metallic AFM tip. The gap localization of the lower mode 2 and its consequence are demonstrated in Supplementary Fig. 6 using numerical simulations. Note that the absence of the lower plasmon mode in the experimental infrared nano-imaging results has also been previously reported in an h-BN separated graphene bilayer structure[29]. Second, the quality factor of the lower plasmon branch is smaller than the higher branch. As shown in the simulation results in Supplementary Fig. 7, the upper plasmon mode 1 can largely dominate the plasmon response when the SWNT plasmon quality factor is moderate.

The experimental and theoretical findings are further corroborated by numerical simulation based on the finite element method (FEM) (see Methods for details). In the numerical simulation, the thin h-BN flake between SWNT and graphene is replaced by air as an approximation to avoid the huge numerical cost, while the dielectric screening effect can be included in the effective permittivity of SWNT. Figure 4a shows the numerically calculated dispersion branches of the coupled modes. The plasmon dispersion is obtained by the Fourier transform amplitude of $Re(E_z)$ at SWNT's bottom surface when an oscillating point dipole source mimicking the tip is used to excite the surface plasmons in the heterostructure. Two dispersion branches in the numerical simulation can be resolved and match well with the theoretical results indicated by the white dashed lines (same as mode 1 and mode 2 in Fig. 3). Figure 4b plots the numerically simulated electric field distribution in the heterostructure at an excitation wavelength of 10.6 μm and a Fermi energy of 0.16 eV. The transverse electric field is largely confined within a few nanometers around the nanotube, which is about 1/1000 of $\lambda_0 = 10.6$ μm. The hybrid plasmon mode thus inherits the strong spatial confinement of Luttinger liquid plasmons in SWNTs. Graphene as a 2D semimetallic sheet further improves the field enhancement at the gap between SWNT and graphene[5].

## Discussion

Further experimental studies are needed to establish conclusively the 1D-2D hybrid plasmons in the SWNT/h-BN/graphene heterostructure. A full characterization of the plasmon dispersion as a function of frequency and momentum, that is $\omega(k)$, will be highly desirable. Systematic study of the hybrid plasmon dispersion in heterostructures with different nanotube diameters and h-BN thicknesses can further elucidate the evolution of the 1D-2D hybrid plasmon as a function of the coupling strength. Experimental realization of these measurements, however, can be challenging. For example, measurements of the plasmon dispersion require a tunable laser over a very broad spectral range away from the phonon bands of both $SiO_2$ and h-BN layers (to ensure strong plasmonic response with a high-quality factor). Such a broadly tunable infrared laser is not available in our current IR-SNOM setup. In addition, the fabrication of clean and gate-tunable SWNT/h-BN/graphene heterostructures with high-quality plasmon modes is quite challenging experimentally, which makes it very difficult to systematically examine the hybridized plasmon oscillations in heterostructures with varying nanotube diameters and h-BN thicknesses. Future improvements on the IR-SNOM apparatus and 1D-2D heterostructure fabrication technique will be needed to realize these measurements.

It should be noted that some coupled systems have been studied before. For instance, graphene was shown to affect the plasmon resonance in a silver nanowire/graphene structure, but only a few percent modulation has been achieved due to the large

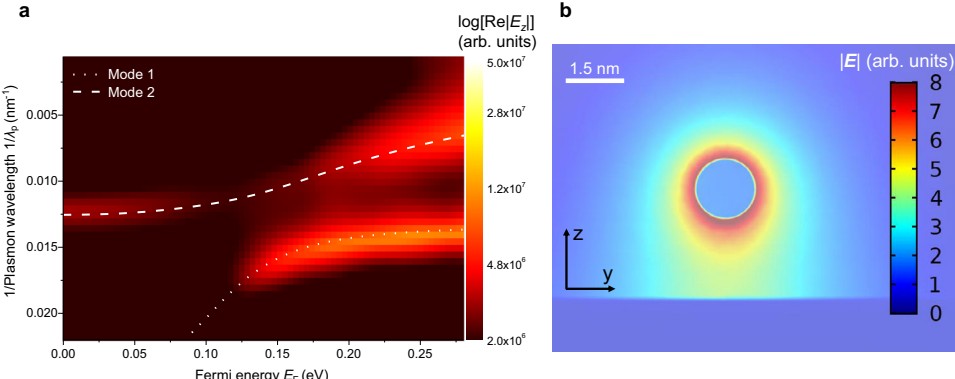

**Fig. 4 Numerical simulation of the plasmon modulation in the heterostructure. a** Fourier transform amplitude of the radial electric field, $|\text{Re}(E_z)|$ at the SWNT surface. $|\text{Re}(E_z)|$ is plotted on a logarithmic scale. Theoretically predicted plasmon dispersion is also shown for comparison (white dashed line: mode 1, white dotted line: mode 2). **b** Simulated electric field amplitude $|E|$ of the hybrid plasmon modes at a plasmonic peak position at an excitation wavelength of 10.6 μm and a Fermi energy of 0.16 eV. The plasmon modes are excited by a z-directed electric point dipole mimicking the tip at the end of the nanotube at a height of 140 nm. The transverse electric field is largely confined within a few nanometers around the nanotube, which is about 1/1000 of $\lambda_O = 10.6$ μm. Source data are provided as a Source Data file.

mode mismatch between graphene plasmons and silver plasmons[21]. Optical responses of other metal/polar crystal heterostructures are constrained to the Reststrahlen bands of the polar crystals and cannot achieve broadband spectral coverage[16]. Although we have focused on a single infrared frequency (28.3 THz) in our study, the coupling effect and plasmon modulation are valid for a broad infrared spectrum where bare SWNT and graphene plasmon momentum can match. The coupled SWNT/h-BN/graphene heterostructure exhibits the combination of broadband response, ultimate confinement limit, and gate tunability, which is superior to other coupled plasmonic systems reported in the literature.

To conclude, we have experimentally realized a gate-tunable plasmonic system composed of low-dimensional vdW materials with mixed dimensionality in a SWNT/h-BN/graphene heterostructure. We observed gate-tunable plasmons in the heterostructure with a 75% plasmon wavelength modulation. Assisted by theoretical modeling and numerical simulations, we suggest that the plasmon modulation may be possible evidence for the hybridization between 1D plasmons in SWNTs and 2D plasmons in graphene. The coupled plasmonic systems with materials of different dimensionality may enable diverse applications in plasmonic nanodevices and light–matter interactions[30–33].

## Methods

**Sample preparation**. SWNTs are directly grown onto $SiO_2$/Si substrates by CVD. Graphene and h-BN flakes are mechanically exfoliated onto $SiO_2$/Si substrates. SWNT/h-BN/Graphene heterostructures are assembled by the standard polymer stamp dry-transfer technique. A polydimethylsiloxane (PDMS) stamp coated with a thin polypropylene carbonate (PPC) film is used to sequentially pick up a thick h-BN flake, a graphene flake, a thin h-BN flake, and SWNTs. Graphene is partially covered by the top thin h-BN flake to allow direct contact of a metal electrode on the exposed area. The PPC thin film together with the heterostructure (SWNT/thin h-BN/graphene/thick h-BN) is peeled from the PDMS stamp, flipped over, and transferred onto a clean $SiO_2$ (285 nm)/Si substrate. The sample is annealed in ultrahigh vacuum (down to $1 \times 10^{-9}$ mbar) at 400 °C for 5 days to remove the PPC film. Standard shadow mask technique is used to fabricate the metal electrodes (100 nm Au/5 nm Cr) contacting the exposed graphene.

**Infrared nano-imaging**. The infrared nano-imaging measurement in this work is accomplished using a home-built scattering-type scanning near-field optical microscope (s-SNOM). To perform infrared nano-imaging, the metallic tip (Nanoandmore HQ:NSC15/Cr-Au-100, tip apex radius ~20 nm) of a tapping mode AFM (Bruker Innova) is illuminated from the side with a p-polarized infrared beam (Access laser L3S $CO_2$ laser, wavelength ~10.6 μm). At the excitation wavelength of 10.6 μm, the plasmonic responses of both SWNT and graphene are

prominent and the effects from the phonon bands of h-BN and $SiO_2$ substrates are negligible. The metallic tip is tapped at a frequency of $\Omega$ ~240 kHz, with an amplitude of ~80 nm. The signal backscattered from the tip apex carries local optical information of the sample and is captured by an MCT detector (Kolmar Technologies KLD-0.1-J1/11/DC) in the far field. To suppress the background scattering from the tip shaft and sample, the detector signal is demodulated at a frequency of $3\Omega$ by a lock-in amplifier (Zurich Instruments HF2LI). By recording the demodulated signal while scanning the sample, near-field images are obtained simultaneously with the topography.

**Algorithmic plasmon wavelength extraction**. We use scikit-image, an open-source image processing library implemented in Python, to determine the plasmon wavelength from the near-field images. The local maxima finding algorithm in scikit-image finds near-field intensity maxima in the experimental images. Local maxima in 2D images of nanotubes are projected on the SWNT axes to avoid potential errors from the nonuniform intensity in the radial direction. We then average the peak-to-peak distances on the SWNT axes to determine the plasmon wavelengths. The uncertainties of the determined plasmon wavelengths are described by the standard deviations of the averaged peak-to-peak distances in plasmon oscillations along SWNTs.

**Numerical simulation**. We use COMSOL Multiphysics to simulate the plasmon dispersion and electric field distribution in the heterostructure. SWNTs are modeled by a metallic cylinder with a radius $R = 0.75$ nm, a length $L = 2$ μm, and a relative permittivity $\varepsilon = -300 + 16.8i$. Graphene is modeled as a 2D conducting sheet with optical conductivity described by the Kubo formula. The graphene relaxation time constant $\tau$ is set to be 100 fs and the temperature $T$ is set to be 300 K. At 10.6 μm, the h-BN substrate has an anisotropic permittivity as $\varepsilon_\perp = 8.343 + 0.023i$ and $\varepsilon_\parallel = 1.933 + 0.006i$. Due to computational limitations originating from the large mismatch between the discretization mesh size (~$10^{-1} \times R$) and the wavelength of the excitation light (10.6 μm), the thin h-BN spacer is replaced by an air gap of 2 nm. A point electric dipole mimicking the tip normal to the graphene surface (that is z-direction) excites the heterostructure at an oscillating frequency of 28.3 THz (~10.6 μm). The radial electric field, $\text{Re}(E_z)$, is taken at the bottom surface of SWNT to avoid dipolar source fields and their diffraction, which can complicate the plasmon dispersion in Fourier transformed profiles.

## Data availability

Data are available within the article and supplementary files. All other data that support the findings of the study are available from the corresponding author upon reasonable request. Source data are provided with this paper[34].

## Code availability

A Python script for the algorithmic plasmon wavelength extraction is available in Zenodo (https://doi.org/10.5281/zenodo.4903983). Codes for electromagnetic calculations are available from the corresponding author upon reasonable request.

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

## Acknowledgements

This work was mainly supported by the Director, Office of Science, Office of Basic Energy Sciences, Materials Sciences, and Engineering Division of the US Department of Energy under Contract No. DE-AC02-05-CH11231 (sp2-Bonded Materials Program KC2207). The SWNT/h-BN/Graphene heterostructure preparation and data analysis were supported by the NSF award 1808635. D.C., F.Wu., and C.Z. acknowledge National Science Foundation for financial support under Grant No. 769K521. K.W. and T.T. acknowledge support from the Elemental Strategy Initiative conducted by the MEXT, Japan, Grant Number JPMXP0112101001, JSPS KAKENHI Grant Number JP20H00354, and the CREST(JPMJCR15F3), JST. S.W. acknowledges the support of the Director's Postdoctoral Fellowship by LDRD program at Los Alamos National Laboratory. S.Y. was supported by the National Research Foundation of Korea (NRF) grants funded by the Ministry of Education (NRF-2017R1A6A3A11034238) and the Ministry of Science and ICT (NRF-2019R1A4A1028121 and CAMM-2014M3A6B3063710).

## Author contributions

F. Wang, S.W., and S.Y. conceived the project. S.W. designed the experiments and performed the infrared nano-imaging measurements. S.W. and S.Z. made the SWNT/h-BN/Graphene heterostructures and devices with help from W.Z., S.K., L.J., M.I.B.U., H.L., S.L., A.Z., E.R., D.W., and Z.Z. D.C. and F. Wu grew the SWNTs under the supervision of C.Z. S.W. proposed the plasmon coupling interpretation. S.Y. established the electromagnetic theory and performed the numerical simulation and performed the quantitative data analysis including the algorithmic extraction of plasmon wavelengths. S.Y., S.W., and F. Wang analyzed the data. S.W., S.Y., and F. Wang prepared the manuscript with input from all authors. K.W. and T.T. provided the h-BN crystals.

## Competing interests

The authors declare no competing interests.
