## [Peer Review File · Nature Communications]

REVIEWER COMMENTS

Reviewer #2 (Remarks to the Author):

This manuscript has been reviewed by multiple reviewers, and their evaluations are all consistent. The authors are presenting some interesting experimental data, which they interpret as a result of strong hybridization of 1D and 2D plasmons. However, the experimental evidence is weak – there are not sufficiently strong experimental results presented that fully support the interpretation. Nonetheless, the authors are not willing to perform additional measurements even though they fully agree with the reviewers that additional measurements are necessary. Instead, they decided to “tone down” their claims by adding the phrase “most likely” in multiple places. In my opinion, this is not sufficient, since the title still explicitly suggests that they have evidence for ‘gate tunable hybrid plasmons in mixed-dimensional van der Waals heterostructures’. Casual readers would mistakenly believe that the authors have evidence for the claim, only because the paper is published in a high-profile journal like Nature Communications. I would therefore suggest that the authors should add “Possible evidence for” at the beginning of the title to be consistent with their toned-down claim. In addition, the lack of solid experimental evidence, i.e., the fact that the claim is just a possibility at this stage, should be clearly stated in the abstract.

Reviewer #3 (Remarks to the Author):

The authors report a combined experimental/theoretical investigation of dimensionally hybrid (1d-2d) plasmons in a van der Waals structure. There are several approximations in their theoretical analysis, and more data would have been welcome, but I view the present work as an initiator of a potentially important subfield. In particular, the tunability of plasmons may have technological impact. In summary, I recommend that this research is published in Nature Communications.

Reviewer #2 (Remarks to the Author):

This manuscript has been reviewed by multiple reviewers, and their evaluations are all consistent. The authors are presenting some interesting experimental data, which they interpret as a result of strong hybridization of 1D and 2D plasmons. However, the experimental evidence is weak – there are not sufficiently strong experimental results presented that fully support the interpretation. Nonetheless, the authors are not willing to perform additional measurements even though they fully agree with the reviewers that additional measurements are necessary. Instead, they decided to “tone down” their claims by adding the phrase “most likely” in multiple places. In my opinion, this is not sufficient, since the title still explicitly suggests that they have evidence for ‘gate tunable hybrid plasmons in mixed-dimensional van der Waals heterostructures’. Casual readers would mistakenly believe that the authors have evidence for the claim, only because the paper is published in a high-profile journal like Nature Communications. I would therefore suggest that the authors should add “Possible evidence for” at the beginning of the title to be consistent with their toned-down claim. In addition, the lack of solid experimental evidence, i.e., the fact that the claim is just a possibility at this stage, should be clearly stated in the abstract.

Reply: We thank the reviewer for his/her comments. We agree with the reviewer’s comment that we need to be careful to state our claims in the revision. Accordingly, we have added “Possible evidence for” at the beginning of the title and in the abstract to avoid confusion as suggested. To be consistent with the changed title and abstract, we also tone down related descriptions and titles of figures in the main text and the supplementary information. The list of changes is appended at the end of this reply letter. We hope our work can promote more studies for dimensionally mixed hybrid plasmons, and the revised version can be satisfactory for publication in Nature Communications.

Reviewer #3 (Remarks to the Author):

The authors report a combined experimental/theoretical investigation of dimensionally hybrid (1d-2d) plasmons in a van der Waals structure. There are several approximations in their theoretical analysis, and more data would have been welcome, but I view the present work as an initiator of a potentially important subfield. In particular, the tunability of plasmons may have technological impact. In summary, I recommend that this research is published in Nature Communications.

Reply: We thank the reviewer for the encouraging comments. In the revision, we clearly state the theoretical and numerical approximations in the main text for clarity. We also tone down our claim by changing the title, the abstract and related descriptions in the main text and supplementary information. The list of changes is appended at the end of this reply letter. We hope our work can be an initiator of tunable hybrid plasmon studies and related subfields.

List of changes

We made the following changes in the revised manuscript to address the points raised by the editor:

1. Modify the title and abstract appropriately

- **Title:** We added “Possible evidence for” in the title following reviewer #2’s suggestion. The title now becomes “*Possible evidence for gate tunable hybrid plasmons in mixed-dimensional van der Waals heterostructures*” in the revised manuscript.
- **Abstract:** the red-colored words are changed in the following sentences:
 - “Here, we report **possible evidence for** gate tunable hybrid plasmons from the dimensionally mixed coupling between one-dimensional (1D) carbon nanotubes and two-dimensional (2D) graphene. In contrast to the carrier density-independent 1D Luttinger liquid plasmons in bare metallic carbon nanotubes, **plasmon wavelengths** in the 1D-2D heterostructure **are modulated by 75%** via electrostatic gating while retaining the high figures of merit of 1D plasmons. We propose **a theoretical model** to describe the **electromagnetic interaction** between plasmons in nanotubes and graphene, **suggesting plasmon hybridization as a possible origin for the observed large plasmon modulation.**”

2. Remove any statements that could be considered to hint at evidence that is not present

- **Title of Figure 2:** the word *hybrid* is removed from the title, “Gate tunable **hybrid** plasmons in a SWNT/h-BN/graphene heterostructure.” (Changes are highlighted by red colors and strikethroughs hereinafter.)
- **Title of Figure 3:** the figure title is changed to “**Possible evidence for** emergence of hybrid plasmons in the SWNT/h-BN/graphene heterostructure.”
- **Title of Figure 4:** the figure title is changed to “Numerical simulation of **the plasmon modulation** in the heterostructure.”
- **Lines 71~75 in page 2:** the following sentences are changed to “In this Letter, we report experimental and theoretical studies **of** plasmons in mixed-dimensional SWNT/h-BN/graphene heterostructures, which serve as an exemplary 1D-2D hybrid plasmonic system. **The results indicate** that the experimentally observed tunable plasmon modes are most likely to be interpreted as the hybrid plasmons due to the coupling between the 1D SWNT plasmons and the 2D graphene plasmons.”
- **Lines 141~142 in page 5:** the word *hybrid* is removed from the sentence, “Fig. 2 presents the main results of infrared nano-imaging of **hybrid** plasmons in a SWNT/h-BN/graphene heterostructure at a set excitation wavelength λ_0 of 10.6 μm .”
- **Lines 179~181 in page 6:** the word *hybrid* is removed from the sentence “The gate tunable **hybrid** plasmons in SWNT M1 and M2 at the positive gate voltage side are

shown in Fig. S8 and shows largely symmetric behavior compared with the negative side.”

- **Lines 196~197 in page 6:** the sentence is changed to “The remarkable plasmon wavelength modulation **represents possible evidence for** the dimensionally mixed coupling between the 1D plasmons in SWNTs and 2D plasmons in graphene.”
- **Lines 212~214 in page 6:** the sentence is changed to “This crossing is split by the optical coupling between 1D SWNTs and 2D graphene when they are stacked together, **implying the possible emergence of** two hybrid plasmon modes.”
- **Line 236~237 in page 7:** the sentence is changed to “A theoretical model is established to better understand the **large plasmon modulation** in this mixed-dimensional heterostructure, **suggesting the likely origin to be the plasmon hybridization.**”
- **Line 279 in page 8:** the sentence is changed to “**We speculate that the** absence of the clear lower plasmon mode in the experimental data is due to a combination of two effects.”
- **Lines 347~350 in page 10:** the sentences in the conclusion are changed to “We observed **gate tunable plasmons in the heterostructure** with a remarkable 75% plasmon wavelength modulation. Assisted by theoretical modeling and numerical simulations, we **suggest that the remarkable modulation may be possible evidence for** the hybridization between 1D plasmons in SWNTs and 2D plasmons in graphene.”
- **Lines 350~352 in page 10:** the word *strongly* is removed from the sentence in the conclusion, “The **strongly** coupled plasmonic systems with materials of different dimensionality may enable diverse applications in plasmonic nanodevices and light-matter interactions³¹⁻³⁴.”
- **Lines 401~403 in page 11:** the word *hybrid* is removed from the sentence in Methods, “The radial electric field, $\text{Re}(E_z)$, is taken at the bottom surface of SWNT to avoid dipolar source fields and their diffraction, which can complicate the **hybrid** plasmon dispersion in Fourier transformed profiles.”
- **Title of Section 4 in SI:** the section title is changed to “**Numerically simulated** hybrid plasmon modes for different gap size h and SWNT diameter d ”
- **Title of Section 8 in SI:** the word *hybrid* is removed from the section title “Gate tunable **hybrid** plasmons in SWNT M1 and M2”
- **Title of Figure S5 in SI:** the figure title is changed to “**Numerically simulated** hybrid plasmon modes for different diameter d and gap size h .”
- **Title of Figure S8 in SI:** the word *hybrid* is removed from the figure title “Gate tunable **hybrid** plasmons in SWNT M1 and M2”

3. Explicitly rule out, at least qualitatively, other interpretations

- **Lines 214~218 in page 6:** the following sentence is added: “Our data show that the

plasmon excitations change gradually over E_F near charge neutrality in our heterostructure (Fig. 2). This behavior rules out an alternative picture based on reflection of graphene plasmons by SWNT because graphene plasmons are strongly suppressed and not observable at the low-doping regime, *i.e.* $|E_F| < 0.12$ eV or $|V_g| < 30$ V.”

4. Emphasize all approximations made in the theoretical analysis

- **Lines 246~247 in page 7:** the red-colored words are added in the sentences, “Using a resonant momentum approximation **by keeping the dominant contribution of the 1D plasmonic field reflected by graphene in its angular spectrum** (see Supplementary Information for details), we can obtain the hybrid plasmon dispersion relation for the SWNT/h-BN/graphene heterostructure,”
- **Lines 255~259 in page 8:** the following sentences are added: “To solve Eq. (1) analytically, we use a permittivity ratio $\epsilon_{in}/\epsilon_{med} = -210+23.52i$ to include the substrate effect effectively. It yields $\lambda_p \sim 80$ nm and $Q \sim 15$ at $E_F = 0$ eV, which are consistent with the experimental finding in Fig. 2 and 3. Further details in the theoretical model and its derivation can be found in section 5 of Supplementary Information.”
- **Lines 302~305 in page 9:** the following sentence is added: “In the numerical simulation, the thin h-BN flake between SWNT and graphene is replaced by an air gap as an approximation to avoid huge numerical cost, while the dielectric screening effect can be included in the effective permittivity of SWNT.”

5. Discuss in detail the experiments that should have been performed to fully support the claimed interpretation

6. Explain why these experiments are out of reach

- **Lines 317~340 in page 9~10:** To address the point 5 and 6, we added the following paragraph:
 - “Further experimental studies are needed to establish conclusively the 1D-2D hybrid plasmons in the SWNT/h-BN/graphene heterostructure. A full characterization of the plasmon dispersion as a function of frequency and momentum, *i.e.* $\omega(k)$, will be highly desirable. Systematic study of the hybrid plasmon dispersion in heterostructures with different nanotube diameters and h-BN thicknesses can further elucidate the evolution of the 1D-2D hybrid plasmon as a function of the coupling strength. Experimental realization of these measurements, however, can be challenging. For example, measurements of the plasmon dispersion require a tunable laser over a very broad spectral range away from the phonon bands of both SiO₂ and h-BN layers (to ensure strong plasmonic response with a high quality factor). Such a broadly tunable infrared laser is not available in our current IR-SNOM setup. In addition, fabrication of clean and gate-tunable SWNT/h-BN/graphene heterostructures with high-quality plasmon modes is quite challenging experimentally, which makes it extremely difficult to

systematically examine the hybridized plasmon oscillations in heterostructures with varying nanotube diameters and h-BN thicknesses. Future improvements on the IR-SNOM apparatus and 1D-2D heterostructure fabrication technique will be needed to realize these measurements.”